# Cytokinin-Facilitated Plant Regeneration of Three *Brachystelma* Species with Different Conservation Status

**DOI:** 10.3390/plants9121657

**Published:** 2020-11-26

**Authors:** Nqobile P. Hlophe, Adeyemi O. Aremu, Karel Doležal, Johannes Van Staden, Jeffrey F. Finnie

**Affiliations:** 1Research Centre for Plant Growth and Development, School of Life Sciences, University of KwaZulu-Natal Pietermaritzburg, Private Bag X01, Scottsville 3209, South Africa; hlophe_n@yahoo.com; 2Indigenous Knowledge Systems (IKS) Centre, Faculty of Natural and Agricultural Sciences, North-West University, Private Bag X2046, Mmabatho 2790, South Africa; 3Laboratory of Growth Regulators, Faculty of Science, Palacký University & Institute of Experimental Botany AS CR, Šlechtitelů 11, CZ-783 71 Olomouc, Czech Republic; karel.dolezal@upol.cz; 4Department of Chemical Biology and Genetics, Centre of the Region Haná for Biotechnological and Agricultural Research, Faculty of Science, Palacký University, Šlechtitelů 27, CZ-783 71 Olomouc, Czech Republic

**Keywords:** Apocynaceae, auxins, conservation, cytokinins, micropropagation, rooting, *meta*-topolin

## Abstract

In Africa and Asia, members of the genus *Brachystelma* are well-known for their diverse uses, especially their medicinal and nutritional values. However, the use of many *Brachystelma* species as a valuable resource is generally accompanied by the concern of over-exploitation attributed to their slow growth and general small size. The aim of the current study was to establish efficient micropropagation protocols for three *Brachystelma* species, namely *Brachystelma ngomense* (endangered), *Brachystelma pulchellum* (vulnerable) and *Brachystelma pygmaeum* (least concern), as a means of ensuring their conservation and survival. This was achieved using nodal segments (~10 mm in length) as the source of explants in the presence of different concentrations of three cytokinins (CK) namely *N*^6^-benzyladenine (BA), isopentenyladenine (iP) and *meta*-topolin riboside (*m*TR), over a period of 6 weeks. The highest (25 µM) concentration of cytokinin treatments typically resulted in significantly higher shoot proliferation. However, each species differed in its response to specific CK: the optimal concentrations were 25 µM *m*TR, 25 µM iP and 25 µM BA for *Brachystelma ngomense*, *Brachystelma pulchellum* and *Brachystelma pygmaeum*, respectively. During the in vitro propagation, both *Brachystelma ngomense* and *Brachystelma pygmaeum* rooted poorly while regenerated *Brachystelma pulchellum* generally lacked roots regardless of the CK treatments. Following pulsing (dipping) treatment of in vitro-regenerated shoots with indole-3-butyric acid (IBA), acclimatization of all three *Brachystelma* species remained extremely limited due to poor rooting ex vitro. To the best of our knowledge, the current protocols provide the first successful report for these *Brachystelma* species. However, further research remains essential to enhance the efficiency of the devised protocol.

## 1. Introduction

The genus *Brachystelma* R. Br. ex Sims is a member of the tribe Ceropegieae belonging to the sub-family Asclepiadoideae of the family Apocynaceae [1]. *Brachystelma* is the second largest genus in the tribe Ceropegieae and occurs in Australia, Southeast Asia, India and sub-Saharan Africa. About 90% of species occur exclusively in sub-Saharan Africa [2]. In South Africa, *Brachystelma* species occur across all the nine provinces [3]. There are 85 *Brachystelma* species on the South African Red List data and 73% are endemic to South Africa. The best species representation is found in the Eastern Cape, with 23 endemic species, and KwaZulu-Natal, with 16 endemic species [3]. Other endemic species occur in other provinces in South Africa [3]. Their distribution is evidence of their variability in habitat preference. Some are found on mountain slopes with up to 3000 mm of annual precipitation, while others are found among rocks in grasslands with up to 200 mm of annual precipitation [2,4,5,6]. One common factor is the requirement for well-drained soils [2]. *Brachystelma* species are generally geophytes, although a small number have fusiform roots instead of a tuber [2,4]. The plants are small (inconspicuous in nature) and herbaceous, generally forming a cluster of deciduous stems with well-developed leaves [4]. The flowers are small and short-lived, and the name “*Brachystelma*”, meaning “short crown”, is associated with the often extremely small corona [4,5].

The tubers of several *Brachystelma* species are eaten raw or prepared by some indigenous groups in Africa, Asia and Australia [2,7,8,9]. Some *Brachystelma* species have also been reported as medicinal herbs against different disease conditions [2,9,10]. Given that *Brachystelma* are being utilized as food and as medicinal herbs, there is a threat to most *Brachystelma* species in nature, mainly due to their slow growth and encroachment on their natural habitats [3,11,12]. The natural regeneration (via seed) process of *Brachystelma* is one that is uncertain and very sparingly documented. Propagation of these geophytes via conventional methods, i.e., cuttings and seedlings, is either absent or not well documented. In addition, conventional methods are generally known to be inefficient compared to micropropagation [13,14,15]. Efficient conservation measures are of importance for the future of the genus *Brachystelma*, especially when their value is yet to be fully explored. Thus, we evaluated the effects of three cytokinins (CKs) on the in vitro propagation of three *Brachystelma* species occurring in South Africa.

## 2. Materials and Methods

### 2.1. Explant Decontamination and Generation of Experimental Material

Whole-plant material of the three *Brachystelma* (*Brachystelma ngomense* R. Br., *Brachystelma pulchellum* R. Br. and *Brachystelma pygmaeum* R. Br.) species were collected from the wild and maintained in a shade house (Figure A1). Based on South African National Biodiversity Institute (SANBI) Red List [3], *Brachystelma ngomense*, *Brachystelma pulchellum* and *Brachystelma pygmaeum* are classified as endangered (EN), vulnerable (VU) and least concern (LC), respectively. These plants were harvested (March and April 2015) from the Botanical Garden of the University of KwaZulu-Natal (UKZN), Pietermaritzburg, South Africa. Voucher specimens (*Brachystelma ngomense*, A. Shuttleworth *335* (NU), *Brachystelma pulchellum*, N. Hlophe 20 (NU) and *Brachystelma pygmaeum*, A. Shuttleworth 322 (NU)) were deposited at the UKZN Bews Herbarium (NU). Young slender stem and tuber explants were excised from the stock plants and thoroughly washed with running tap water.

In the laboratory, the plants were further washed with liquid detergent, followed by thorough rinsing with tap water. In the process of the surface decontamination treatment, the longest tender stem and tuber materials were submerged in 1% (*w*/*v*) Benlate^®^ (Du Pont de Nemours Int., Atholl, Johannesburg, South Africa) for 30 min, followed by a solution of 70% ethanol (*v*/*v*) for 60 s. This process was accompanied with frequent agitation of the solution to ensure maximum contact of plant materials with the sterilant. The plant material was then rinsed three times with distilled water. Thereafter, varying concentrations of two sterilants, namely sodium hypochlorite (NaOCl) (*v*/*v*) and mercuric chloride (HgCl_2_) (*w*/*v*), were used as independent treatments for surface decontamination of the plant materials. The sterilant solutions were supplemented with a few drops of the surfactant Tween 20. The plant material was kept in this solution for varying periods of time, during which the solutions were frequently agitated to allow for optimum contact with the solution. The plant materials were once again thoroughly rinsed three times with sterile distilled water under sterile conditions in a laminar flow bench.

In order to obtain the explants, surface-decontaminated plant materials were further divided into lengths of ~10 mm each and inoculated into culture tubes containing 10 mL of full-strength Murashige and Skoog (MS) basal medium [16], supplemented with 30 g/L sucrose and 0.1 g/L myo-inositol and solidified with 8 g/L bacteriological agar (Oxoid Ltd., Basingstoke, Hampshire, England). The agar was added after the pH of the medium was adjusted to 5.8 using either HCl or NaOH (Sigma-Aldrich, Steinheim, Germany) solutions. The medium was dispensed into culture tubes (100 mm × 25 mm, 40 mL) followed by autoclaving for 20 min at 121 °C and 103 kPa. Sealed cultures were incubated under controlled environmental conditions in a growth room set at 25 ± 2 °C, a 16 h light/8 h dark photoperiod and photosynthetic photon flux (PPF) 40–50 µmol m^−2^ s^−1^ provided by fluorescent tubes (Osram, L58W/640, Munich, Germany). The number of explants for the initial culture of each plant was approximately 50, depending on availability of plant material. After 4 weeks in the culture, the number of sterile explants per treatment was recorded as a percentage. The in vitro-derived aseptic shoots (serving as stem nodal explants) obtained from the sterilization stage were continually sub-cultured until sufficient material was available to conduct subsequent experiments.

### 2.2. In Vitro Shoot Proliferation

Upon obtaining sufficient plant material, we investigated the effects of three CKs, namely *N*^6^-benzyladenine (BA), isopentenyladenine (iP) and *meta*-topolin riboside (*m*TR), on shoot proliferation. We purchased the BA and iP from Sigma–Aldrich (Steinheim, Germany), while *m*TR was prepared in-house as previously described by Doležal et al. [17]. The full-strength MS basal medium was supplemented with 30 g/L sucrose, 0.1 g/L myo-inositol and varying concentrations (1, 5, 10 and 25 µM) of BA, iP and *m*TR. For this experiment, the control was the MS basal medium without CKs. Stem nodal explants were excised to a length of ~10 mm and inoculated into the media in culture tubes. The cultures, 25 replicates per treatment, were incubated using similar environmental conditions as previously described for the bulking-up stage. Data on the number of shoots per explant, the length of the longest shoot, the number of nodal segments, fresh weight, the number of roots and root length were recorded after a 6-week incubation period. This experiment was done simultaneously for all three species and was repeated.

### 2.3. Ex Vitro Rooting and Acclimatization

The in vitro-regenerated shoots (>20 mm) were washed in water to remove traces of agar. Ex vitro rooting was first investigated with the use of a 3 min pulse (dipping) treatment using 492.1 µM (100 mg/L) indole-3-butyric acid (IBA). A second pulse treatment investigation was performed using 492.1 µM of IBA for 3, 12 and 21 min. After treatment with IBA, the shoots were potted in plastic planting trays (45 mm × 15 mm per well) containing a 1:1 (*v*/*v*) vermiculite:sand mixture. The potted shoots were irrigated with quarter-strength 492.1 µM (100 mg/L) of IBA and incubated in a mist house in which high relative humidity (90–100%) was maintained using a high-pressure fog system. The plantlets were kept in the mist house during midwinter under natural 12 h light/12 dark photoperiod conditions for 3 weeks. Thereafter, the plantlets were transferred to a greenhouse with natural temperature (midday PPF of approximately 1000 µmol m^−2^ s^−1^) and natural photoperiodic conditions. The plantlets were watered with tap water every second or third day. Survival rate (%) was monitored once every week for a period of 5 weeks in the greenhouse. The period for which survival rate was monitored was completely dependent on how long the plantlets survived under natural conditions.

### 2.4. Data Analysis

A complete randomized experimental design was used for all experiments. Collected data were analyzed using one-way analysis of variance (ANOVA). In order to establish statistical significances, the mean values were further separated using Duncan’s multiple range test (DMRT) on SPSS for Windows (IBM Corp., Armonk, NY, USA). Significant treatment effects were accepted at *p* ≤ 0.05. Graphic representations were created using SigmaPlot 8.0. (Systat Software, Inc., San Jose, CA, USA, www.systatsoftware.com).

## 3. Results

### 3.1. Decontamination Responses

The use of the 1.75% sodium hypochlorite (NaOCl) treatment for 30 min was found to be most effective, as it resulted in the highest (>60%) decontamination frequency. This treatment was used for the sterilization of all plant material in subsequent experiments (Table 1). The treatment was found to be equally effective for the three *Brachystelma* species.

At this culture initiation stage, stem nodal explants of all three species were observed to have new growth either in the form of shoots or callus; however, a notable number of these explants showed no response (i.e., no elongation or shoot production). Callus was more frequently observed in *Brachystelma pygmaeum* compared with *Brachystelma pulchellum* and *Brachystelma ngomense*. Callus cultures of *Brachystelma pygmaeum* were observed to develop adventitious shoots after a period of 6–8 weeks (Figure A2).

### 3.2. Effect of Cytokinins on In Vitro Shoot Proliferation

The applied CK treatments generally improved shoot production in all three species (Figure 1). The number of shoots produced per explant for *Brachystelma ngomense* were higher with increasing concentrations of BA, iP and *m*TR compared with the control (Figure 1A). The highest shoot production (4.44) was observed at 25 µM *m*TR in *Brachystelma ngomense*. On the other hand, *Brachystelma pulchellum* (Figure 1B) responded in a slightly different manner. There was no particular trend to show the effect of increasing CK concentration with BA and *m*TR. However, iP increased shoot production with increasing concentrations. The highest shoot production (2.04 ± 0.20) was observed at 25 µM iP, which was significantly higher than the control. For *Brachystelma pygmaeum* (Figure 1C), all treatments displayed a significantly higher shoot production per explant compared with the control, with the exception of 1 µM iP and *m*TR. Treatment with BA resulted in a significant increase in shoot production with increasing concentrations, while iP and *m*TR had a fluctuating response. In all three treatments, the higher concentrations resulted in better shoot production. The highest shoot production (2.57) was observed with the 25 µM BA treatment.

In the current study, the shoot lengths and number of nodal segments of the three *Brachystelma* species generally had no significant difference between CK-treated explants and the control (Figure 2 and Figure 3); however, a few exceptions were observed. The shoot length in *Brachystelma ngomense* (Figure 2A) was significantly increased by 5 µM iP. The number of nodal segments in *Brachystelma pulchellum* (Figure 3B) and *Brachystelma pygmaeum* (Figure 3C) were significantly increased by 5 µM *m*TR and 5 µM BA, respectively. To a certain degree, the application of CKs partly enhanced fresh weight in all three species (Figure 4A). The effect of BA, iP and *m*TR on the fresh weight of *Brachystelma ngomense* (Figure 4A) was observed to be significantly higher in comparison with the control, with some exceptions. The highest fresh weight (0.18 g) was observed at 25 µM *m*TR. For *Brachystelma pulchellum* (Figure 4B), treatment with BA had no significant effect on fresh weight, while specific concentrations of iP and *m*TR treatments significantly increased fresh weight compared with the control. The highest fresh weight (0.10 g) was observed at 5 µM *m*TR. In *Brachystelma pygmaeum* (Figure 4C), fresh weight was significantly increased by all CK treatments compared with the control, with the exception of 1 µM BA and 1 µM *m*TR. The highest fresh weight (0.23 g) was achieved at 5 µM BA and 5 µM iP.

Although the primary aim of this particular experiment was to observe the effect of CKs on *Brachystelma* shoot growth parameters, rooting was also observed in in vitro-regenerated *Brachystelma ngomense* and *Brachystelma pygmaeum* shoots (Figure 5 and Figure 6). The highest number of roots, as well as the longest root length, was observed in the control. On the other hand, the treatments with BA, iP and *m*TR had an antagonistic effect on rooting in all three *Brachystelma* species. In *Brachystelma pygmaeum*, however, the lower concentrations of BA were found to have the least inhibiting effect on rooting (Figure 5B). No roots were induced in CK-regenerated *Brachystelma pulchellum* and the control. Nevertheless, the poorly rooted or rootless shoots were transferred to ex vitro conditions. The shoots of *Brachystelma ngomense*, *Brachystelma pulchellum* and *Brachystelma pygmaeum*, however, did not form “spontaneous” roots under ex vitro conditions. A survival rate of 0% was observed after 2 weeks in the mist house and 1 week in the greenhouse.

In vitro abnormalities observed in this study include hyperhydricity (Figure A2) and “dwarfing” (Figure A3) in the form of an abundance of small leaves on a short node section. Hyperhydricity mainly affected adventitious shoots derived from the callus. Shoots derived from nodal explants were seldom affected by hyperhydricity. No further morphological differences were observed between the parent plants and regenerants.

### 3.3. Effect of Pulsing (Dipping) Treatment on Ex Vitro Rooting and Acclimatization.

Pulse (dipping) treatment of regenerated shoots with IBA (100 mg/L = 492.1 µM) for 3 min showed an improved greenhouse survival compared with shoots previously potted directly after in vitro root induction. During acclimatization, the shoots derived from this experiment had an extended survival period beyond 10 weeks as opposed to the 7-week survival of the in vitro root induction treatments. This extension can be attributed to the ex vitro root induction that was observed by the fourth week under greenhouse conditions. The survival rate after 4 weeks was 42, 35 and 30% for *Brachystelma ngomense*, *Brachystelma pulchellum* and *Brachystelma pygmaeum*, respectively. By the end of the 10^th^ week, the survival rate was 5%, 0% and 3% for *Brachystelma ngomense*, *Brachystelma pulchellum* and *Brachystelma pygmaeum*, respectively.

The few surviving plants maintained a healthy green appearance with no obvious morphological abnormalities for both the above- and belowground organs (Figure A3). The healthy green appearance was perceived as an indication of functional photosynthetic capacity. Subsequently, *Brachystelma pygmaeum* shoots derived in vitro were used to test the effect of pulse treatments with 100 mg/L of IBA at different time intervals (3, 12 and 21 min). By the end of 8 weeks, the survival rate was 5% for 3, 12 and 21 min, while there was no survival (0%) for the control. Thus, the exposure time had no effect on ex vitro root induction and the overall survival of shoots remained low.

## 4. Discussion

Contaminants in plant tissue culture can cause huge economic losses by either making the explant unfit for sub-culture or resulting in its death [18,19]. Even though the 1.75% NaOCl treatment for 30 min was found to be the most effective for surface sterilization, a notable number (±40%) of explants were lost. Among the likely reasons contributing to contamination in this particular study were the aphids, which were feeding on the stock plants (Figure A2). Plant feeders or pests, especially aphids, do not only contribute to surface contamination but also internal contamination which is an even bigger challenge when attempting to obtain aseptic plants [18]. Thus, for the purpose of optimizing the decontamination success rate, it is necessary to ensure that stock plants are well protected prior to use so as to avoid huge losses [20].

At the culture initiation stage, stem nodal explants of all three species were observed to have new growth either in the form of shoots or callus; however, a notable number of these explants showed no response. Generally, the lack of response from meristematic tissue may be due to bud dormancy or failure in stem elongation [14]. In the current study, some nodal explants taken from various parts of the stem showed no growth response (i.e., no elongation or shoot production). Thus, it is suggested that the reason for this lack of growth response is not based on a specific position on the stem but is likely due to the physiological state of the plant material. The physiological state of an explant determines its biological response; therefore, some explants are more responsive than others at a given time. For instance, callus was frequently observed in *Brachystelma pygmaeum* compared with *Brachystelma pulchellum* and *Brachystelma ngomense*. Callus cultures of *Brachystelma pygmaeum* were observed to develop adventitious shoots after a period of 6–8 weeks (Figure A2). On the other hand, tuber explants of *Brachystelma pygmaeum* did not show any indication of new growth during the four weeks in culture. It is likely that there is an absence of meristematic potential in *Brachystelma* tuber tissue. There have been reports that totipotency is highly likely to be influenced by the presence of plant growth regulators when it does not occur spontaneously [21,22,23]. Failure to perform subsequent experiments on *Brachystelma pygmaeum* tubers and to evaluate tubers of the other two *Brachystelma* species was due to the unavailability of starting plant material. Secondary explants obtained from initiated cultures were used for bulking up plant material.

An increase in shoot numbers as a response to CK treatments has been observed in various micropropagated plants [24]. This observation is thought to reflect the effect of increasing concentrations of CKs during micropropagation [25,26]. Even though the results of the current study have not shown a particularly significant difference among the different CKs (BA, iP and *m*TR), other studies have pointed out the typical differences that result from treatment with CKs belonging to different groups, mainly the aromatic CKs and isoprenoids [27,28,29]. Differences are also observed even within the CK groups [17,30]. Cytokinins, or plant growth regulators in general, have different mechanisms of stimulation, depending on factors such as plant species and plant organs. In vitro proliferation rate and biomass accumulation are typically the primary indicators of a CK treatment’s stimulation effects [31]. Aromatic CKs, including BA and *m*TR, are generally known to have a greater influence on developmental processes, mainly those related to morphogenesis and senescence, hereas isoprenoids, including iP, are suggested to be more involved in growth processes concerning the continuation of the cell cycle [32,33,34]. *N*^6^-benzyladenine (BA), a synthetic CK, remains the most widely used exogenously applied aromatic CK in commercial micropropagation due to its effectiveness and affordability, but BA is also known to have deleterious effects on in vitro cultures [17,24,35]. Isopentenyladenine (iP), a natural isoprenoid CK, is also reported to cause physiological abnormalities and to have weak activity in the in vitro propagation of some plant species, especially in comparison with BA [28,36]. Thus, other aromatic CKs such as the *meta*-topolins, which are hydroxylated 6-benzyladenine derivatives, have been identified as possible alternatives [37].

*Meta*-topolins, including *m*TR, have generally been observed to not only result in improved shoot proliferation but also cause minimal physiological abnormality [30,38,39,40,41,42,43]. For instance, Bairu et al. [44] obtained higher shoot multiplication rates from a number of *meta*-topolin derivatives at different concentrations compared with BA. On the other hand, some studies have reported lower multiplication rates from *meta*-topolins compared with BA [30,45,46]. Meanwhile, some studies show no significant difference between treatments [47,48]. Thus, it is evident that responses to plant growth regulator treatments remain species-specific. Generally, the better performance of the *meta*-topolins has been attributed to their structural advantage when compared with BA [24,39,49]. The advantage of the topolin structure is due to the presence of hydroxyl (OH) groups on the benzyl ring, which, during CK metabolism, increase the chances of the formation of *O*-glycosides instead of the deleterious *N*-glycosides found in BA-treated plantlets [50]. Furthermore, the *O*-glycoside metabolites are considered stable only at times when they are not required by the plant but are rapidly converted to active CK bases when required [24]. This allows the continuous availability of physiologically active CK over an extended period of time, thus resulting in enhanced shoot formation in vitro [24]. On the other hand, the chemical and biochemical stability of *N*-glucoside is the reason for their implication in the deleterious effect of BA in plant tissue culture, which may extend to the acclimatization stage [39].

In vitro rooting is controlled by both endogenous and exogenous auxins in plant tissue [39]. Regardless of the absence of exogenous auxins in the medium, cultured plantlets are generally able to produce roots, particularly in a medium without plant growth regulators. However, a medium supplemented with high concentrations of CKs has inhibitory effects. A study by Valero-Aracama et al. [35] reported that higher concentrations of BA and *m*TR inhibited rooting in *Uniola paniculata* (sea oats). Similar observations have been made in the current study, where treatment with higher concentrations of BA inhibited in vitro rooting (Figure 5B). On the other hand, iP and *m*TR, irrespective of the concentration, inhibited in vitro rooting. Treatment with BA is typically known to reduce acclimatization competence [35,51]; however, in the current study, acclimatization competence was lacking, regardless of the CK treatment.

The decrease in survival in this current study is primarily attributed to the poor rooting, which is commonly observed in micropropagation systems [52,53]. Some studies have shown that increasing exposure time promotes rooting [54]. In a study by Phulwaria et al. [55], pulse treatment with IBA (100 mg/L) for 3 min was particularly effective in ex vitro root induction for *Ceropegia bulbosa*. Thereafter, 100% rooting was observed, which was followed by successful hardening and transfer to the field [55]. Even though the pulse treatment used in this study did not yield a positive result, ex vitro rooting is known to be more advantageous in comparison with in vitro rooting, especially for plants that are difficult to root [53,55].

Concerning acclimatization incompetence, there are a couple of factors that have been reported as possible contributors. Light stress is one of the possible contributing reasons for this short-lived survival, as transfer of plantlets to in vivo conditions of higher light intensity is known to have an overwhelming effect [53,56]. Humidity is another factor that changes drastically, thus affecting plantlet survival [56]. Successful establishment of plantlets ex vitro, as a concluding step, is crucial in micropropagation because many biotechnological applications are dependent on plant regeneration efficiency [19,57]. Poor survival during acclimatization is often attributed to the heterotrophic mode of nutrition under which the plantlets develop morpho-physiological disorders such as poor control of water loss [52,53,57]. However, *Brachystelma* has been observed to be a naturally fragile group of plants, which might very well be a factor contributing to their poor performance in root development and acclimatization. Perhaps a study involving seed germination may yield better results and also shed some light on the rooting process of *Brachystelma* species.

## 5. Conclusions

The study demonstrated that *Brachystelma* species can be manipulated under in vitro culture conditions, though not optimally at present. The manipulation of the culture media with various parameters (in this case, different CK types) and the use of controlled environmental conditions had moderate effects on the development of micropropagation protocols for the three *Brachystelma* species. Further acclimatization of *Brachystelma ngomense*, *Brachystelma pulchellum* and *Brachystelma pygmaeum* was extremely limited due to the poor rooting that led to short-lived survival under greenhouse conditions. Concerns about a decline in the wild of *Brachystelma* populations call for an improvement of the available standard micropropagation protocol which can be specifically used in their conservation. Once established, these protocols can also be used with other biotechnological applications involving *Brachystelma*. The results obtained in the current study provide a stepping-stone for subsequent research towards enhanced rapid clonal propagation of *Brachystelma* species, which is needed, especially in the absence of efficient alternative conventional propagation techniques.

## Figures and Tables

**Figure 1 plants-09-01657-f001:**
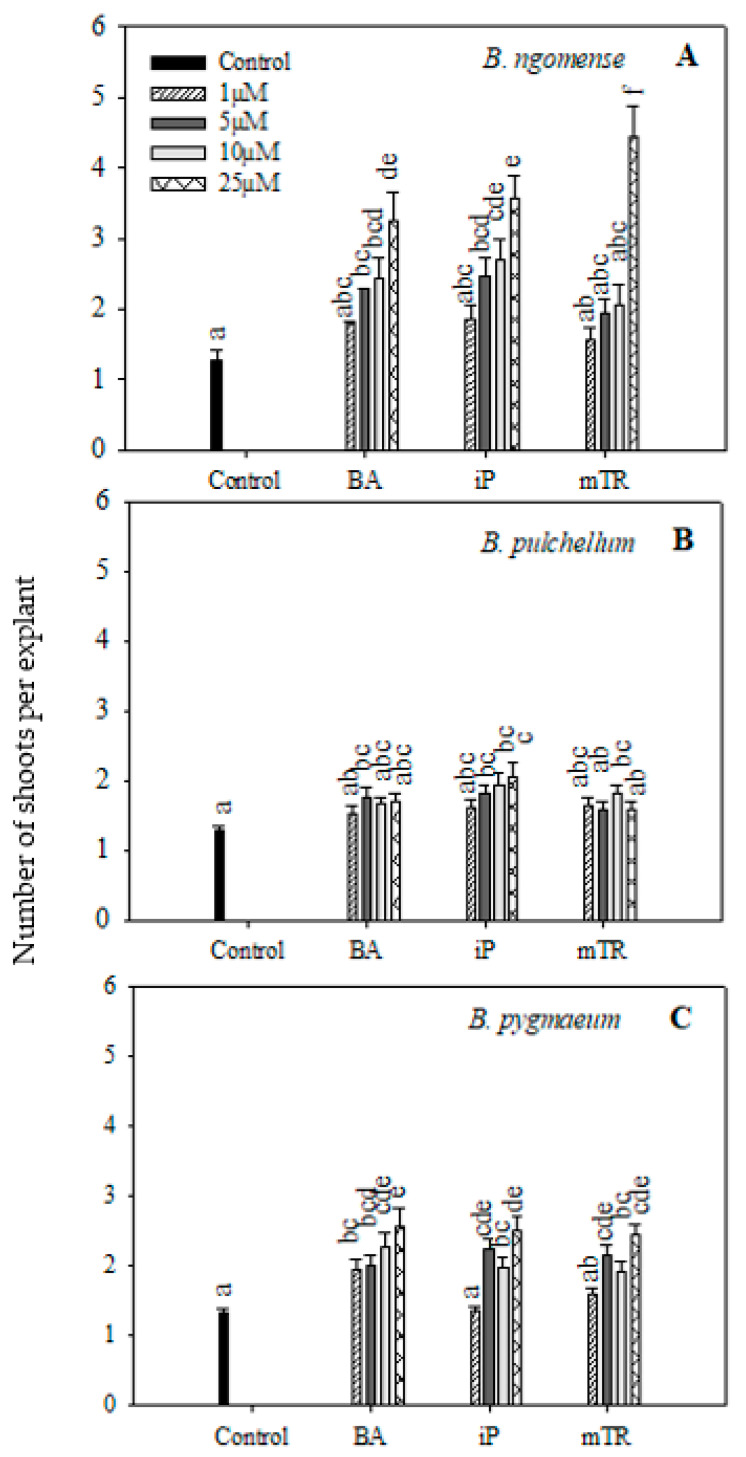
Effect of varying concentrations of cytokinins on shoot production in (**A**) *Brachystelma ngomense*, (**B**) *Brachystelma pulchellum* and (**C**) *Brachystelma pygmaeum* after six weeks in culture. In each graph, different letter(s) on the bars show significant differences according to Duncan’s multiple range test (DMRT) (*p* ≤ 0.05). *N*^6^-benzyladenine (BA), isopentenyladenine (iP) and *meta*-topolin riboside (*m*TR).

**Figure 2 plants-09-01657-f002:**
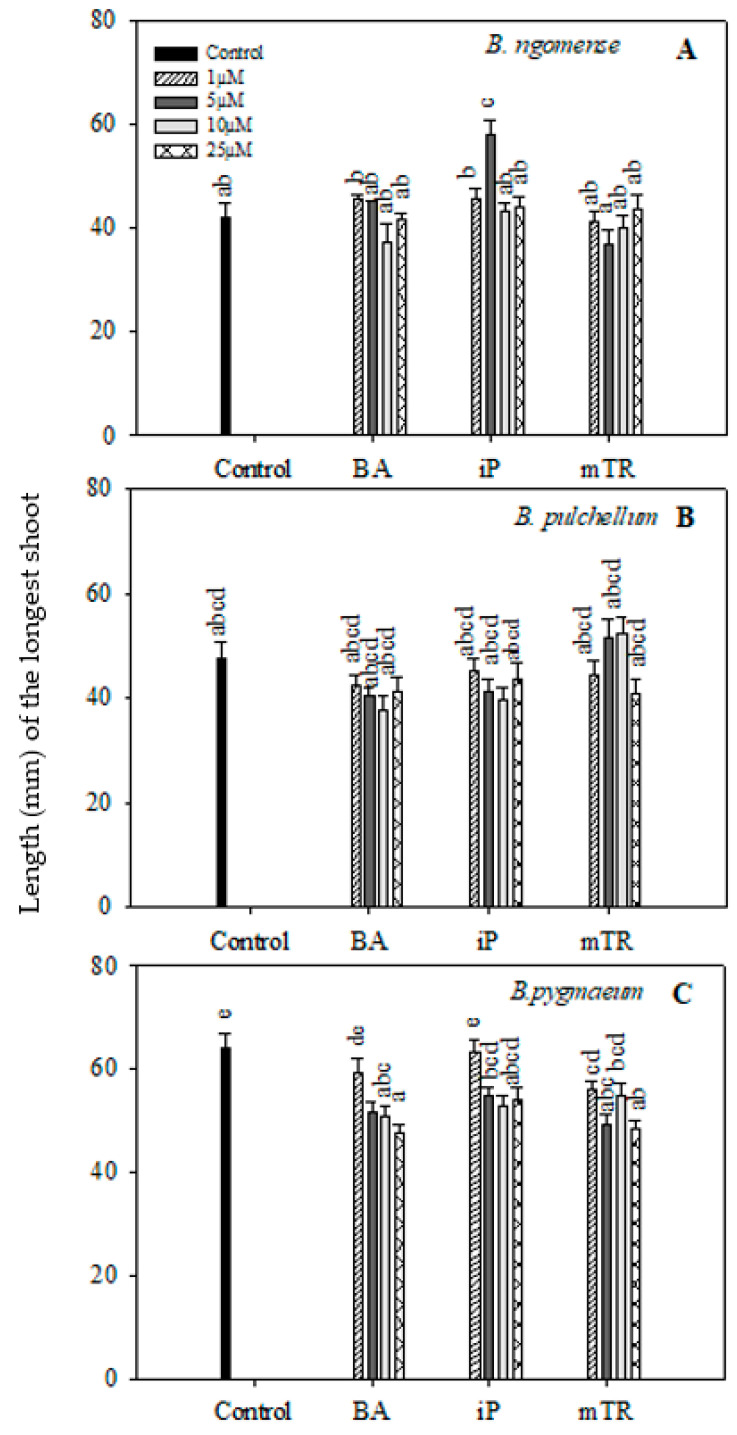
Effect of varying concentrations of cytokinins on the shoot length (mm) of (**A**) *Brachystelma ngomense*, (**B**) *Brachystelma pulchellum* and (**C**) *Brachystelma pygmaeum* after six weeks in culture. In each graph, different letter(s) on the bars show significant differences according to Duncan’s multiple range test (DMRT) (*p* ≤ 0.05). *N*^6^-benzyladenine (BA), isopentenyladenine (iP) and *meta*-topolin riboside (*m*TR).

**Figure 3 plants-09-01657-f003:**
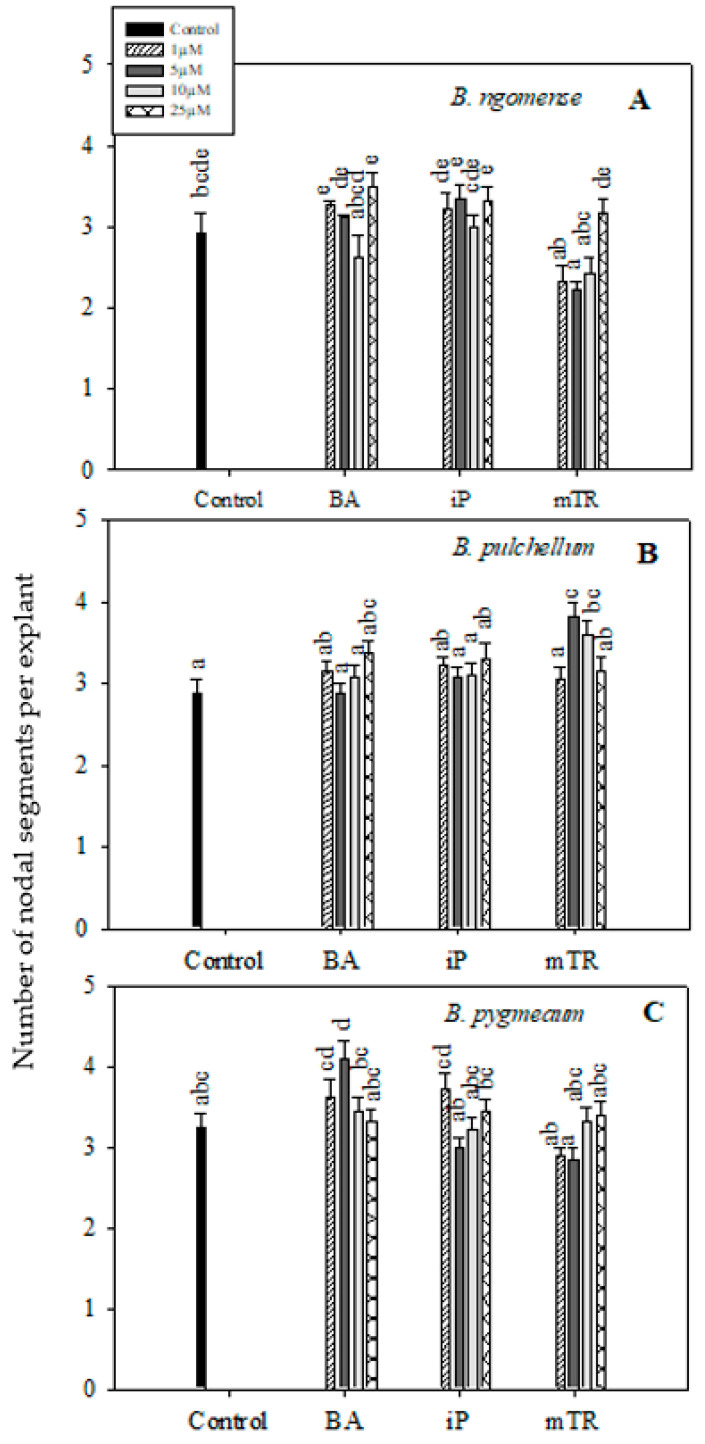
Effect of varying concentrations of cytokinins on the number of nodal segments in (**A**) *Brachystelma ngomense*, (**B**) *Brachystelma pulchellum* and (**C**) *Brachystelma pygmaeum* after six weeks in culture. In each graph, different letter(s) on the bars show significant differences according to Duncan’s multiple range test (DMRT) (*p* ≤ 0.05). *N*^6^-benzyladenine (BA), isopentenyladenine (iP) and *meta*-topolin riboside (*m*TR).

**Figure 4 plants-09-01657-f004:**
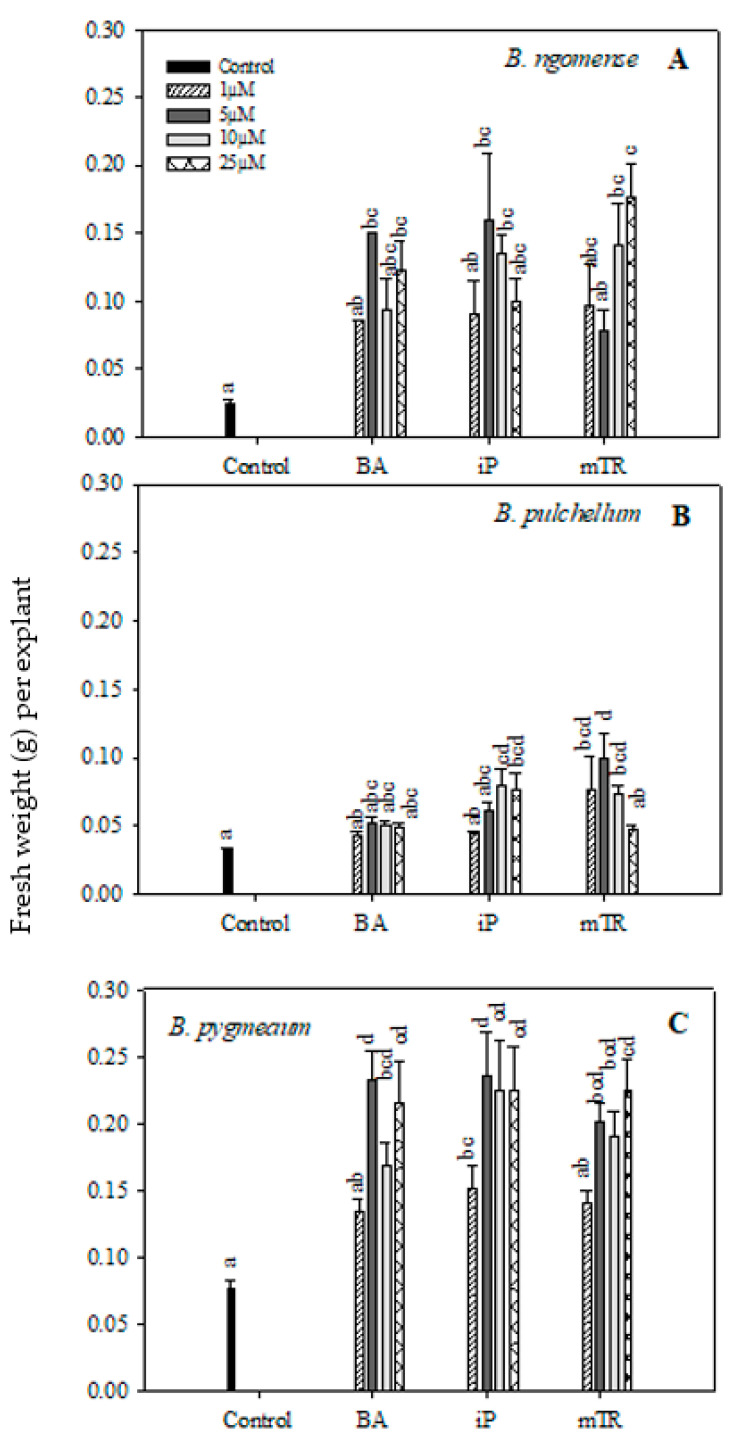
Effect of varying concentrations of cytokinins on fresh weight (g) in (**A**) *Brachystelma ngomense*, (**B**) *Brachystelma pulchellum* and (**C**) *Brachystelma pygmaeum* after six weeks in culture. In each graph, letter(s) on the bars show significant differences according to Duncan’s multiple range test (DMRT) (*p* ≤ 0.05). *N*^6^-benzyladenine (BA), isopentenyladenine (iP) and *meta*-topolin riboside (*m*TR).

**Figure 5 plants-09-01657-f005:**
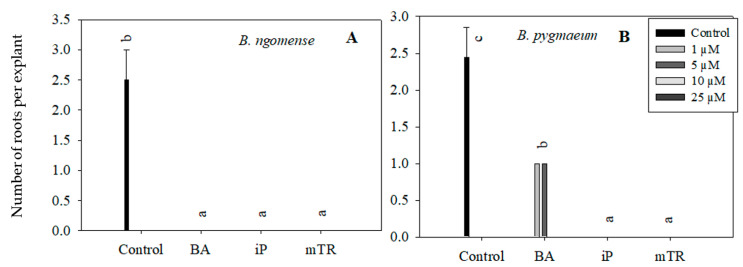
Effect of varying concentrations of cytokinins on the number of roots in (**A**) *Brachystelma ngomense* and (**B**) *Brachystelma pygmaeum* after six weeks in culture. In each graph, different letter(s) on the bars show significant differences according to Duncan’s multiple range test (DMRT) (*p* ≤ 0.05). *N*^6^-benzyladenine (BA), isopentenyladenine (iP) and *meta*-topolin riboside (*m*TR).

**Figure 6 plants-09-01657-f006:**
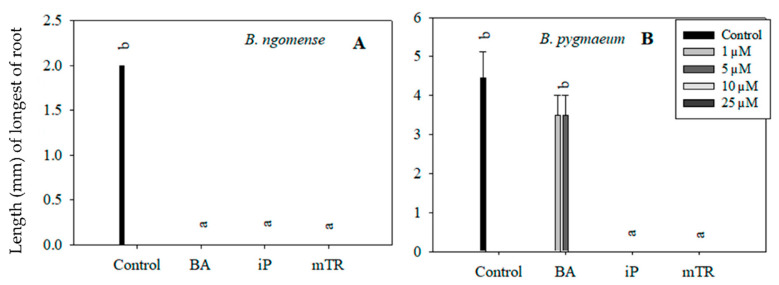
Effect of varying concentrations of cytokinins on the root length (mm) in (**A**) *Brachystelma ngomense* and (**B**) *Brachystelma pygmaeum* after six weeks in culture. In each graph, different letter(s) on the bars show significant differences according to Duncan’s multiple range test (DMRT) (*p* ≤ 0.05). *N*^6^-benzyladenine (BA), isopentenyladenine (iP) and *meta*-topolin riboside (*m*TR).

**Table 1 plants-09-01657-t001:** Surface decontamination of *Brachystelma pygmaeum* plant material using mercuric chloride (HgCl_2_) and sodium hypochlorite (NaOCl) for different time durations.

Treatment	Duration (min)	Explant	Decontamination Success (%)
HgCl_2_ (0.1%)	5	Nodal	0
HgCl_2_ (0.1%)	10	Nodal	9
HgCl_2_ (0.2%)	30	Tuber	0
NaOCl (1.75%)	30	Nodal	67
NaOCl (3%)	10	Nodal	20

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
