# Peer review of "Cytokinin-Facilitated Plant Regeneration of Three Brachystelma Species with Different Conservation Status"

_plants, 2020, doi:10.3390/plants9121657_

Round 1
Reviewer 1 Report
The comments are as follows:
M&M
- please include size of the explants, additional photograph showing both used explants would be valuable
- lines 114-118 - please insert more details on the nodes, were they on the main stem or branches too, or might be only on the longest shoot?. How many repetitions was applied?
- line 121- what means pulse treatment, please explain in details.
Results
- - Authors tested different chemicals and timing of disinfection (lines 88-94), however in the results (lines 142-146) they do not give any details. If you have pointed that out in methods, please give full results of your investigations, or either remove this information from the manuscript.
- Line 195 – bushiness is not a precise term, as it suggests appearance of additional branching. Looking at the submitted pictures, I suggest to change it to dwarfing (shortening of the stem and decrease in the leaf size). Please consider that thought and correct accordingly.
- Figs. Y axis description should be more precise: in example Fig 1 no of shoots per explant, Fig 2 Length of the longest shoot etc.
- In M&M section Authors provided information that they tested 2 types of explants (lines 81-82), but results are not specified in that accordance. If the results are presented just for stem explants exclusively, what about the other type (tuber explants). If the results are presented jointly, please specify that and change the results and discussion accordingly.
- Line 353 – kind confusing, since only cytokines has been tested.
Reviewer 2 Report
I have the following points to be considered for further improvement of the manuscript:
- Authors should include proper permit numbers from the authorized agency while working with endangered plant species, especially work that entails removing plants from the wild.
- Authors should specify the manufacturer's name for each chemical used in the study.
- Authors should describe in detail the sterilization treatment for percentage, time and explants used in the study (nodal segments or tubers), providing detailed information in each step.
- There are 25 replicates, but how many plants are in each replication? Explain whether the experiment was done once or repeated.
- Suggestion to use the acronym for Isopentenyladenine: 2iP
- Authors should consider doing in vitro rooting using various auxins, not cytokinins. Generally, you will not see rooting due to cytokinin supplements in the media.
- Authors should also consider various levels of IBA for ex vitro experiments, otherwise justify the use of one level of IBA in the experiments.
- Authors should justify the use of this protocol for micropropagation as the survival rate is very low, with an explanation of future directions to improve the survival rate.
Reviewer 3 Report
The manucript titled „Cytokinin-facilitated plant regeneration of three Brachystelma species with different conservation status” is interesting and the plant species are worth of investigation although some minor corrections required, see below.
Introduction section:
No need to put again the numer of cytokinines and the numer of species in the brackets: „Thus, we evaluated the effects of three (3) cytokinins (CKs) on the in vitro-propagation of three (3) Brachystelma species occurring in South Africa.” – delete it, please.
The expression „bulking” should be deleted from the subsection heading „Explant decontamination and bulking of experimental material”
This sentence needs grammar corrections: „Where there were statistical significances, mean values were further separated using Duncan’s multiple range test (DMRT) on SPSS for Windows (IBM SPSS Statistics 24, USA).”
Reviewer 4 Report
The authors presented a study on three Brachystelma species listed on SANBI’s Red List as endangered, vulnerable and least concern. The obtained results may be valuable for the micropropagation of those species if the method of rooting is fine-tuned in the future.
- First of my concern in the number of repetitions. Did authors perform only 2 biological repeats (25 explants per each)? The plant material for the repeats were collected from the same plant or different? When the repeats were done in time (or when the material for each repeat was collected and prepared)?
- I would be interested in more specific results from the decontamination response then presented in the paragraph 3.1. Could the authors provide plot or table with the results? Moreover the presence of aphids feeding on the plants may influenced the response in the culture. All of the used material in the study has been attacked by the aphides?
- The symbol of μM has been changed for some strange signs in the all six plot’s legends.
- In the Fig.2.B and C the significant differences are marked only for three variables not all four. Why?
- Why the authors check the influence of CK on rooting? Did they check auxins which are usually used for rooting initiation in in vitro cultures, dark and half nutrition media? That would enhance the rooting and acclimatization which is crucial in the plant micropropagation protocol.
- How many regenerants/ after what supplementation during the whole procedure did authors obtain?
- If I understand well the authors would recommend for the micropropagation of ngomense and B. pygmaeum the supplementation of 25 μM BA, iP or mTR. Could the authors provide some photos and phenotype description/observable characteristics in comparison to control? I am wonder if such high concentration of CK may result in abnormalities which disqualify such supplementation from practical usage.
- According to paragraph 3.3 and observed in vitro abnormalities did the authors checked or cloud check the presence of somaclonal variation in the regenerants?
Round 2
Reviewer 1 Report
Not all issues were adressed in the revised version.
Please insert more details on the nodes, were they on the main stem or branches too, or might be only on the longest shoot?.
How many repetitions was applied in the experiment?
Reviewer 2 Report
n/a
Reviewer 4 Report
The authors provided valuable changes and responses for all my questions and considerations.
